# Barley *SIX-ROWED SPIKE3* encodes a putative Jumonji C-type H3K9me2/me3 demethylase that represses lateral spikelet fertility

Hazel Bull[1,2], M. Cristina Casao[2], Monika Zwirek[2], Andrew J. Flavell[3], William T.B. Thomas[2], Wenbin Guo[3,4], Runxuan Zhang[4], Paulo Rapazote-Flores[4], Stylianos Kyriakidis[3], Joanne Russell[2], Arnis Druka[2], Sarah M. McKim[3] & Robbie Waugh [2,3]

The barley inflorescence (spike) comprises a multi-noded central stalk (rachis) with tri-partite clusters of uni-floretted spikelets attached alternately along its length. Relative fertility of lateral spikelets within each cluster leads to spikes with two or six rows of grain, or an intermediate morphology. Understanding the mechanisms controlling this key developmental step could provide novel solutions to enhanced grain yield. Classical genetic studies identified five major *SIX-ROWED SPIKE* (*VRS*) genes, with four now known to encode transcription factors. Here we identify and characterise the remaining major *VRS* gene, *VRS3*, as encoding a putative Jumonji C-type H3K9me2/me3 demethylase, a regulator of chromatin state. Exploring the expression network modulated by *VRS3* reveals specific interactions, both with other *VRS* genes and genes involved in stress, hormone and sugar metabolism. We show that combining a *vrs3* mutant allele with natural six-rowed alleles of *VRS1* and *VRS5* leads to increased lateral grain size and greater grain uniformity.

[1] James Hutton Limited, The James Hutton Institute, Invergowrie, Dundee DD2 5DA, Scotland. [2] Cell and Molecular Sciences, The James Hutton Institute, Invergowrie, Dundee DD2 5DA, Scotland. [3] Division of Plant Sciences, School of Life Sciences, The University of Dundee at The James Hutton Institute, Invergowrie, Dundee DD2 5DA, Scotland. [4] Information and Computational Sciences, The James Hutton Institute, Invergowrie, Dundee DD2 5DA, Scotland. Hazel Bull and M. Cristina Casao contributed equally to this work. Correspondence and requests for materials should be addressed to S.M.M. (email: s.mckim@dundee.ac.uk) or to R.W. (email: robbie.waugh@hutton.ac.uk)

All wild barleys have two-rowed spikes with each rachis node bearing a single grain from a fertile central spikelet flanked by two sterile lateral spikelets. Six-rowed types, where both central and lateral spikelets bear grain, arose post-domestication some 12,000–8000 years before present through selection of spontaneous recessive six-rowed variants. Despite the potential to produce up to three times as many grain per spike, a commensurate reduction in spike-bearing stems (tillers) per plant and smaller lateral grain size ultimately results in comparable yields between both morphological types[1] (https://cereals.ahdb. org.uk/varieties/ahdb-recommended-lists/winter-barley-201718. aspx). Extensive classical genetic studies have identified up to 11 independent loci (variously called HEXASTICHON (HEX), SIX-ROWED SPIKE (VRS) and INTERMEDIUM (INT)) associated with complete or partial changes in the fertility of the lateral spikelets[2]. Of the characterised row-type genes, VRS1 (syn. HvHOX1) encodes a basic helix-loop-helix (bHLH) transcriptional activator[3, 4] that inhibits the development of fertile lateral spikelets. Mutations in VRS1 alone are both necessary and sufficient to generate a full six-rowed phenotype, leading to VRS1 being considered the key molecular gatekeeper in the control of lateral spikelet fertility. VRS2[5], VRS4[6] and VRS5/INT-C[7] encode homologues of SHORT INTERNODES (SHI)[8], LATERAL ORGAN BOUNDARY (LOB)[9] and TEOSINTE BRANCHED1/ CYCLOIDEA/PROLIFERATING CELL NUCLEAR ANTIGEN FACTOR 1 (TB1)[10] transcription factors, respectively, whose functions appear partially conserved. Recessive mutant alleles exhibit varying degrees of six-rowed phenotypic penetrance and cause so-called intermedium phenotypes. The genetic data indicate that alleles of VRS1 and VRS5 are epistatic[7], and expression

analyses suggest that VRS4 acts early to promote VRS1 expression[6].

Here we show that the remaining unidentified major VRS gene, VRS3, encodes a putative Jumonji C-type (JMJC) H3K9me2/me3 demethylase whose loss of function is associated with gains in lateral spikelet fertility. Deep gene expression analyses revealed that VRS3 promotes expression of all other VRS genes throughout spikelet development, and that VRS3 function modulates networks associated with sugar and hormone metabolism, and stress signalling. Although natural VRS3 alleles are not strictly associated with row-type, combining mutant vrs3 alleles with natural six-rowed VRS1 (vrs1.a) and VRS5 (Int-c.a) alleles results in increased lateral grain size and uniformity.

## Results

### vrs3 mutants show gains in lateral spikelet fertility.

Mutants of VRS3 (vrs3 syn. int-a) in a two-rowed cv. Bowman background (BW419, int-a *BC6) typically show two-rowed architecture in the lower portion of the spike and six-rowed architecture in the upper part (Fig. 1a, b). Lateral spikelet fertility appears coincident with the development of awns on the lateral spikelets (Fig. 1c–e and Supplementary Note 1). Penetrance of these characteristics is environmentally and genetically sensitive, and can be accompanied by supplementary florets, spikelets and ectopic awns on the palea (Fig. 1f–h and Supplementary Note 1). Bowman and a second near isogenic line (NIL), BW902(vrs3.f *BC6)[11], show striking differences in early lateral spikelet development. When Bowman spikes are 2–3 mm long (late lemma primordium stage), the appearance and elongation of awn initials on the central spikelet marks the beginning of the awn primordium (AP) stage and the end of lateral spikelet development, with the laterals subsequently remaining small and undifferentiated (Fig. 1i, j). At the same developmental stage, BW902(vrs3.f) lateral spikelets clearly increase in size with enlarged lemmas enclosing differentiating floral organs (Fig. 1j). Despite these differences in inflorescence morphology, no significant difference in days to spike emergence was observed in individuals from F2 populations segregating for mutant/wild-type alleles at the vrs3 locus.

### VRS3 encodes a putative JMJC H3K9me2/3 demethylase.

Genetic analyses of the two NILs carrying independent vrs3 mutations (BW419(int-a.1) and BW902(vrs3.f)) located VRS3 on the short arm of chromosome 1H[11]. We mapped vrs3 in an F2 segregating population (BW419(int-a.1)*Barke) to a 10.9 cM interval using a 384 SNP Illumina BeadXpress genotyping platform and then systematically refined this interval genetically using additional SNP markers to a 2.2 cM interval containing 22 gene models[12, 13] that exhibited conserved synteny with an orthologous region on rice chromosome 10 (Fig. 2a, b and Supplementary Table 1). Within this interval lies the rice gene LOC_OS10g42690, encoding a JMJC H3K9me2/3 demethylase, which when mutated, perturbs rice inflorescence morphology[14]. Sequence-based phylogeny (Supplementary Note 1 and Supplementary Fig. 1) suggested that HvJMJ706 was a likely functional orthologue of OsJMJ706 and therefore a promising candidate for VRS3. In support of this hypothesis, sequencing HvJMJ706 in BW419(int-a.1) and BW902(vrs3.f) revealed frameshift mutations within conserved C5HC2 zinc finger and JmjN functional domains, respectively, suggesting that sequence changes in HvJMJ706 may represent vrs3 loss of function alleles. We confirmed the identity of HvJMJ706 as VRS3 by sequencing a further thirty-two row-type mutants, generated using different mutagens and previously shown to be alleles of vrs3 by genetic complementation[15]. Thirty-one alleles contained disruptive lesions in HvJMJ706 (26 independent; Fig. 2c). The vrs3.f allele produced

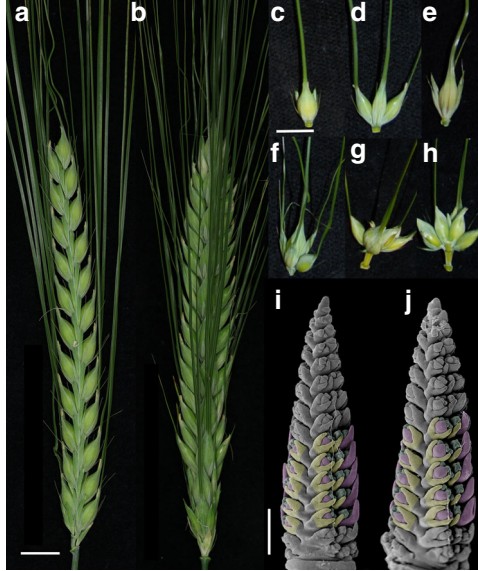

**Fig. 1** The vrs3 phenotype. **a** Spike of the two-rowed cv. Bowman. **b** Spike of the vrs3 NIL BW419(int-a.1). Scale bar in **a** applies to **b**, 1 cm. **c** Bowman spikelet triplet with fertile central and infertile lateral spikelets. **d–h** Spikelet triplets from a homozygous vrs3 individual from the BW419(int-a.1) *Bowman F2 population. **d** Six-rowed spikelet triplet. **e** Spikelet triplet with fertile central spikelet and infertile, pointed lateral spikelets. **f–h** Spikelet triplet with additional fertile spikelets and awned palea. **g** Adaxial and **h** abaxial views of the same spikelet triplet showing the formation of additional florets within the central spikelet. Scale bar in **c** applies to **d–h**, 1 cm. **i, j** Scanning electron microscopy of developing spikes of **i** Bowman and **j** vrs3 NIL BW902(vrs3.f). Developing lemmas are highlighted in pink, anthers in green and glumes in yellow. Scale bar in **i** applies to **j**, 500 μm

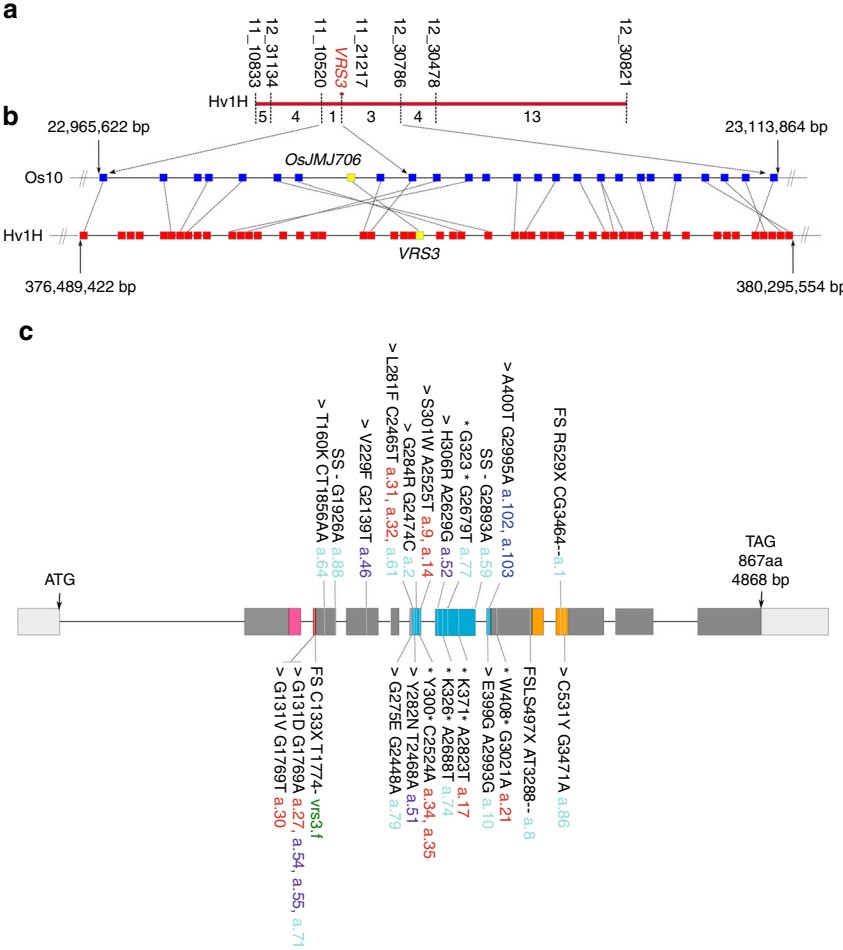

**Fig. 2** Identification of *VRS3*. **a** The refined genetic mapping interval of the *vrs3* locus showing the number of recombinants between each marker. **b** Conserved synteny between orthologous regions on the physical maps of rice chromosome 10 and barley chromosome 1H within the genetic mapping interval. Rice gene models are shown in dark blue and barley in red. The yellow square represents the *VRS3* gene candidate. **c** *VRS3* gene model with regions encoding three putative functional domains JmjN (pink), JmjC (turquoise) and C5HC2 zinc-finger (orange). The 5′ and 3′untranslated regions are shown in pale grey. Positions of the mutant alleles are relative to the start codon (ATG). Allele colours represent the respective genetic background of the induced mutation: turquoise (Bonus), green (Hakata 2), dark blue (Hege), red (Foma), and purple (Kristina). Types of mutation are coded as: > non-synonymous substitution, *premature stop, FS, frameshift, SS, splice site

the most truncated protein and the most severe phenotype. An analysis of growth and development of the mutant allelic series revealed that the phenotypic penetrance of *vrs3* was influenced by both genetic background and environment. We observed that genetically distinct lines possessing identical non-synonymous substitutions (*int-a*.27 (Foma), *int-a*.71 (Bonus) and *int-a*.54 (Kristina)), segregating progenies from bi-parental mapping crosses and the same lines grown in the field or glasshouse displayed variable inflorescence phenotypes (Supplementary Note 1).

**Two major *VRS3* haplotypes exist in European germplasm.** To assess whether natural alleles of *VRS3* were associated with the different spike row-types, as demonstrated previously for *VRS1* and *VRS5*[3, 7, 16], we sequenced *VRS3* across 22 representative two-rowed and six-rowed, winter and spring planted European cultivars, resolving two major haplotypes, *Vrs3*.w *and Vrs3*.x. These are distinguished by six SNPs in complete linkage disequilibrium. One SNP generates a non-synonymous substitution (serine/asparagine substitution in exon 2), which PROVEAN analysis[17] suggests is neutral. KASP marker analysis in a collection of 482 European cultivars revealed that, unlike *VRS1* and *VRS5*, and similar to *VRS2* and *VRS4*, *Vrs3*.w and *Vrs3*.x

haplotypes are not wholly-associated with alternative row-types (Fig. 3a). However, the observed allele frequencies in winter and spring genotypes may explain the observed germplasm-dependent appearance of row-type associations with *VRS3* in GWAS experiments[7, 18] and may reflect selection for a linked spring or winter genepool-specific character.

**VRS3 does not appear to be a target of directional selection.** To explore the wider natural diversity of *VRS3*, we mined an exome capture data set from a geo-referenced panel of 86 wild and 132 locally adapted landraces (53 two-rowed and 79 six-rowed)[19]. We identified a total of 64 SNPs. Of 48 exonic SNPs, 21 generated synonymous and 27 non-synonymous substitutions. Only one non-synonymous change occurred within a conserved functional domain: a T > A substitution (MAF 0.92%) that resulted in a grass-conserved asparagine to lysine substitution within the JmjC domain that once again PROVEAN[17] suggests is neutral. Median-joining network analysis clustered the 64 SNPs into 46 haplotypes with 37 unique to wild germplasm, two to the two-rowed, four to the six-rowed landraces and three that were mixed (Fig. 3b). *Vrs3.w* and *Vrs3.x* haplotypes are most closely related to network haplotypes 45 and 46, respectively. These two haplotypes

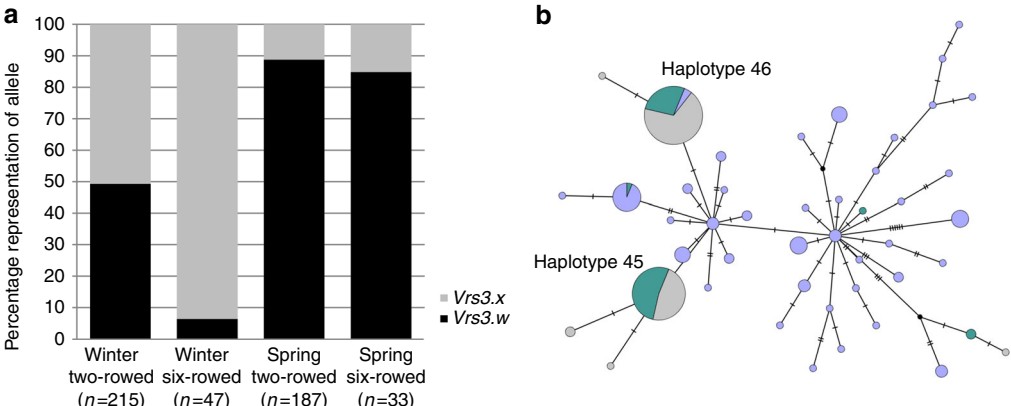

**Fig. 3** The allelic diversity of *VRS3* across wild, landrace and cultivated barley germplasm. **a** Frequency of *Vrs3*.w and *Vrs3*.x alleles across the major classification divisions (winter-sown, spring-sown, two-rowed and six-rowed) within cultivated barley. **b** Median joining network illustrating the occurrence and relationships between different *Vrs3* haplotypes across wild species (purple), two-rowed landrace (green) and six-rowed landrace (grey) germplasm. Tick marks across the connecting bars represent the number of polymorphisms distinguishing haplotype groups. The size of the circle is proportional to the number of representatives within the allelic class. *Vrs3*.w and *Vrs3*.x are renamed haplotype 45 and 46 because three SNPs at the 5′ and 3′ end of the exome capture alignments were missing from the alignment

derive from haplotype 7, represented by three wild accessions that are each geo-referenced to Israel, suggesting that the cultivated haplotypes most likely originate from Israeli wild barleys. There was no obvious correlation between *VRS3* alleles, row-type or geography within the landraces, suggesting *VRS3* has not been subject to directional selection (Fig. 3b and Supplementary Fig. 2).

### VRS3 promotes expression of row-type transcription factors.

The available RNA-seq data (http://camel.hutton.ac.uk/barleyGenes_JLOC2) indicated that *VRS3* is expressed throughout the developing barley plant (Supplementary Note 1 and Supplementary Fig. 3). Using qRT-PCR, we observed a 3-fold change in expression in *Vrs3*.w in very young 1 mm Bowman spikes (triple mound stage) compared to later stages (Supplementary Fig. 4a, b), suggesting an early developmental role, although we did not observe any morphological phenotypes at this stage. However, RNA in situ hybridization at the early lemma primordium stage (spike length 2 mm) revealed *Vrs3*.w expression in developing glumes of lateral spikelets, and in the central and distal rachis vasculature at later stages (Fig. 4a–f and Supplementary Fig. 4).

To understand more about the molecular basis of *VRS3* suppression of lateral spikelet fertility, we next performed highly replicated, comparative RNA-seq ($n = 8$/stage/genotype) using mRNA from inflorescences of Bowman and BW902(*vrs3*.f). We chose two developmental stages: AP (spike length 4–6 mm), when spike patterning begins to differ between genotypes; and white anther (WA; spike length 9–11 mm), when floret primordia reach maximal number. After removing all 107 differentially expressed (DE) genes that lay within the genetic introgression containing *vrs3*.f to avoid confounding effects, we found a total of 364 DE genes ($|\log_2$ fold change (LFC)$| \geq 0.5$; adjusted $P < 0.05$) in at least one of the stages, with 162 upregulated and 202 downregulated in BW902(*vrs3*.f) (Fig. 4g, h, Supplementary Data 1 and Supplementary Note 1). As our results suggest that *VRS3* is likely a functional orthologue of *OsJMJ706*, we hypothesised that VRS3 may remove the same repressive H3K9me2/3 chromatin mark as the rice enzyme. Thus, we focused on DE genes that were downregulated in BW902(*vrs3*.f). VRS1 and VRS5 were repressed in AP (*VRS1*, LFC = −2.039, adjusted $P = 7.01 \times 10^{-9}$; *VRS5*, LFC = −1.329, adjusted $P = 2.36 \times 10^{-6}$) and WA (*VRS1*, LFC = −1.512, adjusted $P = 5.56 \times 10^{-8}$; *VRS5*, LFC = −1.204,

adjusted $P = 4.56 \times 10^{-5}$) spikes. *VRS2* and *VRS4* were downregulated in BW902(*vrs3*.f) AP inflorescences (*VRS2*, LFC = −0.655, adjusted $P = 0.019$; *VRS4*, LFC = −0.698, adjusted $P = 2.33 \times 10^{-5}$) but not in WA spikes. These observations alone may explain why *vrs3* mutants produce a partial six-rowed phenotype (Fig. 4i–m).

### VRS3 modulates gene expression.

The RNA-seq data set also revealed striking trends in sugar metabolism and hormone pathways consistent with the increased lateral growth in BW902(*vrs3*.f) and the localisation of *VRS3* expression across spike development (Fig. 4a–g). Trehalose-6-phosphate phosphatase (*T6PP*, HORVU6Hr1G074960) was repressed in AP spikes (LFC = −1.041; adjusted $P = 1.84 \times 10^{-4}$, Supplementary Fig. 5d), as reported in *vrs4* mutants[6], reflecting perturbed T6P homeostasis during inflorescence development and growth[20, 21]. A bHLH transcription factor (HORVU4Hr1G075320, LFC = −0.910, adjusted $P = 3.45 \times 10^{-4}$ (AP); LFC = −1.175, adjusted $P = 2.18 \times 10^{-7}$ (WA)) with homology to maize *PTF1*[22] implicated in controlling soluble sucrose content was also repressed. Three *SWEET* genes (also known as saliva or *MtN3*) that encode transmembrane proteins involved in sugar transport, were significantly DE in BW902(*vrs3*.f) spikes at WA stage: HORVU5Hr1G076770 and HORVU7Hr1G054710 were downregulated (LFC = −1.106, adjusted $P = 5.03 \times 10^{-4}$ and LFC = −1.147, adjusted $P = 0.004$, respectively) while HORVU3Hr1G107780 (LFC = 0.736; adjusted $P = 0.002$) was upregulated. These genes participate in a wide range of the biological processes in plants such as host-pathogen interactions, reproductive development, senescence and abiotic stress responses[23]. All three are highly expressed in developing inflorescence and caryopsis tissues in barley (http://camel.hutton.ac.uk/barleyGenes_JLOC2). HORVU5Hr1G076770 and HORVU7Hr1G054710 are orthologs of *Xa13/Os8N3/OsSWEET11* in rice, where suppressed expression causes reduced fertility or sterility due to compromised microspore development[23, 24]. Here, downregulation was associated with increased fertility of BW902(*vrs3*.f) spikes; however, the *SWEET* gene HORVU3Hr1G107780 was significantly upregulated at the same WA stage, consistent with the effect of *SWEET* genes in the rice inflorescence[25]. The contrasting regulation we observe here suggests that different *SWEET* genes may play different roles in determining barley spike architecture and pollen fertility. Downregulation of cytokinin oxidase/dehydrogenase (*HvCKX*) (revealed by

qRT-PCR, Supplementary Fig. 5d), an enzyme that mediates cytokinin degradation, is consistent with increased lateral outgrowth and grain number associated with increased cytokinin levels[5, 26]. A homolog of *LONELYGUY-LIKE* (*LOG-LIKE*), a cytokinin-activating enzyme necessary to preserve meristem

activity[27, 28] was upregulated in both stages (HORVU4Hr1G079860, LFC = 2.005, adjusted $P = 9.78 \times 10^{-10}$ (AP); LFC = 1.920, adjusted $P = 8.44 \times 10^{-10}$ (WA); Supplementary Fig. 5d), similar to observations in other cereal inflorescence architecture mutants[6, 27]. Other highly upregulated genes in

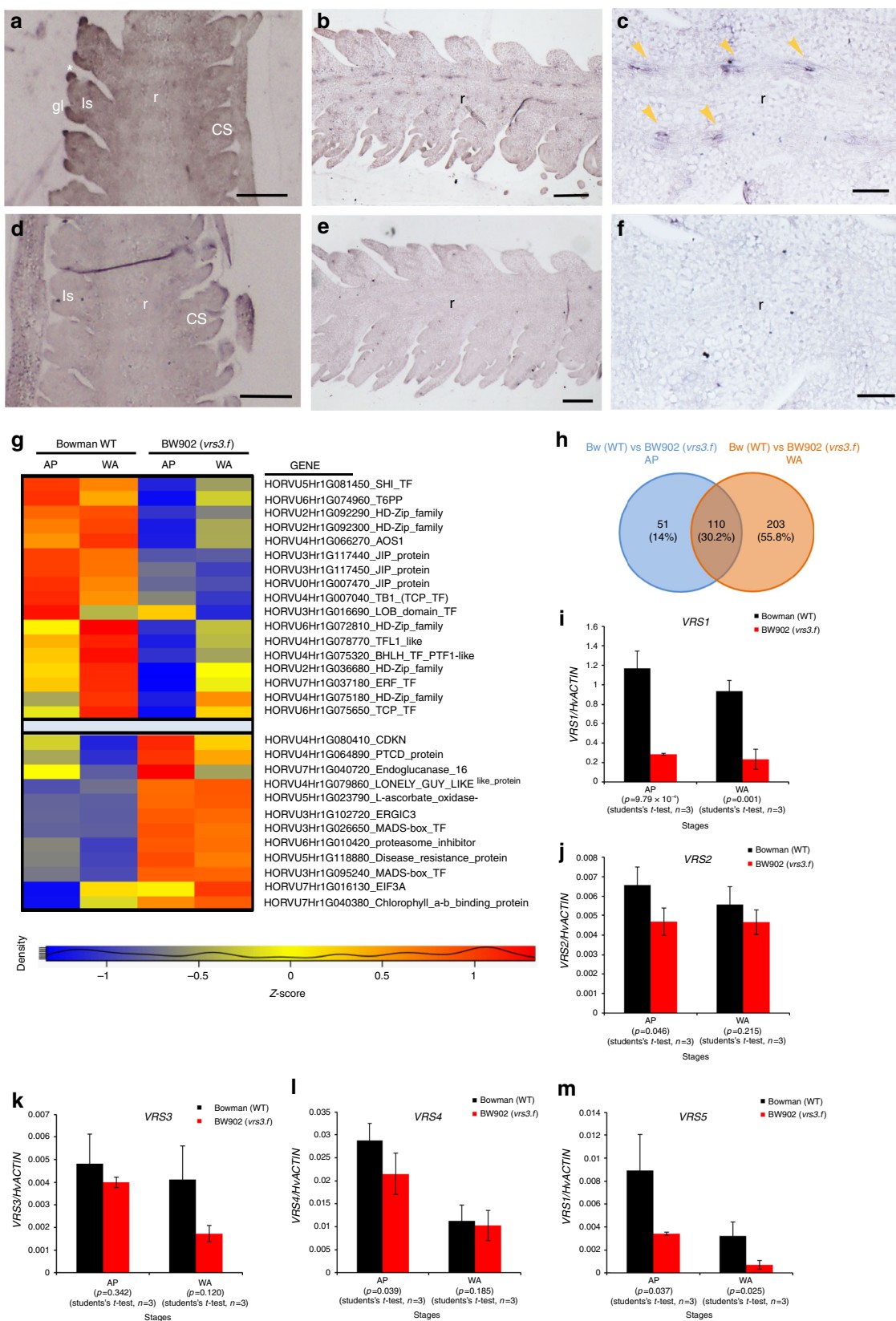

BW902(vrs3.f) were associated with increased growth (e.g., EIF3A[29]), cell proliferation (e.g., ERGIC3[30]) and embryo development (e.g., PPR proteins[31]).

We were intrigued to find stress and defence signalling to be the only statistically enriched biological function in our DE gene set (adjusted $P = 2.69 \times 10^{-2}$, Supplementary Data 1). Inspection of the over-represented transcripts revealed multiple thionins and peroxidases downregulated in BW902(vrs3.f) (Supplementary Note 1). This was supported by downregulation of allene oxide synthase 1 (AOS1, HORVU4Hr1G066270), the first enzyme in the lipoxygenase pathway of jasmonate (JA) biosynthesis[32], a classic plant hormone involved in fertility, wounding and defense[33] and a number of jasmonate-induced proteins (JIPs[34]). In rice, loss of JA biosynthesis/signalling is also associated with extra floral meristem and floral organ formation[35], potentially relevant to the ectopic awn/lemma and floret phenotypes observed in BW902(vrs3.f). Overall, our comparative RNA-seq data provides two levels of interpretation: (1) VRS3 may regulate the expression of VRS1, VRS5 and other row-type genes in order to maintain the two-rowed spike form and, (2) DE genes in BW902(vrs3.f) reveal roles for sugar and cytokinin pathways, but also an intriguing link to JA signalling associated with the control of lateral spikelet fertility.

**vrs3 improves grain uniformity in six-rowed germplasm**. As only the natural alleles Vrs3.w and Vrs3.x are present within elite cultivated six-row barley germplasm, we tested whether introducing mutant alleles of VRS3 offered any potential for crop improvement[36]. We crossed BW902(vrs3.f) and BW419(int-a.1) into the six-row cultivar Morex (vrs1.a, Int-c.a) and used diagnostic molecular markers to identify all allelic classes of progenies. Comparing central and lateral grain sizes from lines homozygous for six-row alleles of VRS1(vrs1.a) and VRS5(Int-c.a), and homozygous for either wild type (Vrs3.w) or mutant VRS3 (vrs3) revealed that vrs3 offered a significant improvement in lateral grain width and area (Fig. 5, Supplementary Note 1 and Supplementary Fig. 6). Importantly, the lateral to central grain size ratio significantly increased from 89 to 93% ($P = 0.035$, ANOVA, $n = 6$ (662), $n = 12(666)$), improving a critical processing character.

## Discussion

Our data suggest that VRS3 is a putative JMJC H3K9me2/me3 demethylase required for barley floral organ development by regulating expression of floral development genes. Assuming this functional assignment is correct, our observations are consistent with a model where VRS3 maintains a locally permissive H3K9 methylation state in cells of the developing inflorescence, which directly or indirectly facilitates transcription of the major VRS genes that together coordinate the maintenance of lateral spikelet sterility. Deleterious mutations, such as vrs3.f, would induce a phenotypic response primarily by maintaining repressive chromatin around the major VRS genes, restricting transcription and partially phenocopying recessive natural vrs mutations that confer a six-row phenotype. It remains unclear why introducing vrs3 into an otherwise six-row background (vrs1.a Int-c.a) improves the lateral to central grain size ratio. Transcriptional responses in vrs3.f indicate that either subsequently, or co-operatively, plant hormone balance, especially cytokinins and JA, sugar metabolism and stress responses are involved in regulating lateral spikelet fertility. Its putative functional role may help explain the varied genetic and environmental penetrance of the mutant phenotype.

## Methods

**Germplasm and phenotyping**. Germplasm is listed in Supplementary Table 2. The vrs3 (syn = int−a) mutant allelic series was obtained from the Nordic genebank. Bonus, Hege, Bowman, Foma, Morex and Kristina cultivars were from the JHI Barley collection. Hakata 2 was obtained from Okayama University (accession J807). VRS3 was genotyped across 220 spring and 262 winter European cultivars from Cockram et al.[37] and Tondelli et al.[38]. Two Bowman near-isogenic line alleles for vrs3, BW419 (int-a.1, an X-ray induced mutant in Bonus*BC6) and BW902 (vrs3.f, gamma-ray-induced mutant of Hakata2* BC6) were crossed to Bowman, Barke and Morex cultivars. F2 plants deriving from BW902*Bowman, BW419*Bowman, BW902*Barke and BW419*Barke crosses were grown in 23 cm pots within a polytunnel and phenotyped at four positions within the main spike (top, middle and two lowest spikelet nodes) for row-type, grain fill, awn length, additional florets and spikelets, and awned paleas. BW419*Morex and BW902*Morex F2 plants used in row-type gene pyramiding experiments were grown in 23 cm pots under long-day glasshouse conditions (16 h light/8 h dark, at 18 °C day and 14 °C night). Plants for SEM, in situ hybridisation, qPCR and RNA-seq were grown in glasshouses under the same long-day conditions.

**Identification of VRS3**. BW419*Barke F2 individuals ($n = 108$) were phenotyped, DNA extracted and genotyped using an Illumina BeadXpress SNP platform[39]. Further polymorphic markers in the region were identified from B-OPA SNPs used previously to genotype Barke and Bonus[40]. KASP allele-specific PCR assays (LGC Genomics) designed to individual informative SNP's directly from the Illumina OPA manifest files were performed according to the manufacturer's guidelines. Genotyping and KASP primers are listed in the Supplementary Data 2. Map distances were calculated using JoinMap3 (Kyazma). Local conservation of synteny between barley and rice was established using Strudel[41].

**Phylogenetic analysis**. Reciprocal blast of the VRS3 predicted protein sequence against the plant Phytozome database 12 identified 18 sequences from the Gramineae (E score of 0.0) predicted to contain all three functional domains and sharing at least 50% identity. BLAST of the wheat survey sequence[42] identified three orthologues on chromosomes 1A, 1B and 1D. BLAST against the barley predicted proteins[12, 13] identified a paralogue of VRS3, MLOC_53868.1, on chromosome 6H. BLAST of this sequence against the wheat survey sequence[42] again identified homologous sequence on chromosome 1A and 1D; however, no reliable sequence could be established for chromosome 1B. Protein sequences were aligned using MUSCLE[43]. The BMGE algorithm[44] removed 740 phylogenetically uninformative characters across the aligned sequences prior to phylogenetic analysis. Phylogenetic analysis was performed in MEGA6[45] using the 'maximum likelihood nearest-neighbour interchange method'. A bootstrap consensus tree was inferred from 1000 replicates (Supplementary Data 3).

**Fig. 4** Expression analyses. **a–f** In situ RNA hybridization of VRS3 in two-rowed barley cv. Bowman at lemma primordium (LP) (spike length 2 mm) and awn primordium (AP) stage (spike length 5 mm). **a–c** Longitudinal sections hybridised with antisense VRS3 probe (**a**) at LP stage showing VRS3 signal in the developing glumes (gl) of the lateral spikelet (ls, in middle plane) and potential signal (*) from the glume of the adjacent central spikelet (CS) and at (**b**) AP stage showing expression in the rachis (r, yellow arrowheads) and (**c**) at higher magnification. **d–f** Equivalent longitudinal sections hybridised with sense probe. Scale bars, 100 μm (**a**, **b**), 200 μm (**c**, **d**), 50 μm (**e**, **f**). **g** Heat map of differentially expressed (DE) genes implicated in regulating spike development and meristem identity. Legend indicates gene expression abundances (Z scores) across the different developmental stages AP (5 mm) and white anther (WA) (10 mm) in BW902(vrs3.f) and Bowman ($n = 8$ bioreps per genotype). **h** Venn diagram showing the overlap of DE genes between Bowman (WT) and BW902(vrs3.f) in AP (5 mm) and WA (10 mm) spikes. Numbers represent the DE genes genetically located outside of the vrs3.f introgression in BW902(vrs3.f). **i–m** Average gene transcript levels determined by quantitative RT-PCR in Bowman WT (black) and BW902(vrs3.f) (red) in AP and WA spikes; **i** VRS1; **j** VRS2; **k** VRS3; **l** VRS4; **m** VRS5. HvActin was used for normalization. x-axis shows the inflorescence developmental stages in which the RNA-seq experiment was performed. y-axis shows the relative expression level based on ΔCt (cycle threshold) calculation. Mean ± S.E of three biological replicates is shown. P values were calculated based on Student's t tests (two-tailed)

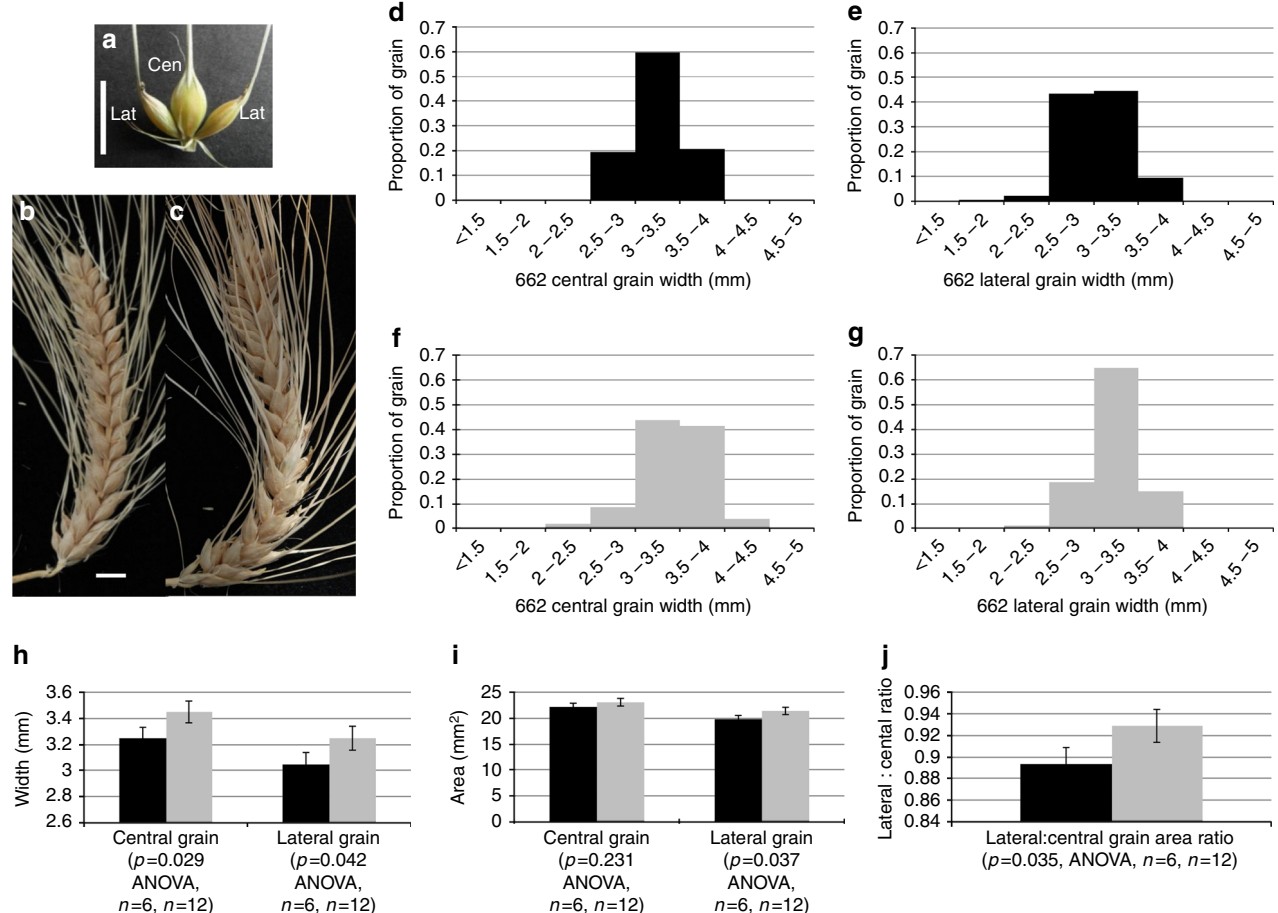

**Fig. 5** Impact of *vrs3* on grain size and uniformity in the six-rowed spike. **a** A typical six-rowed barley spikelet triplet (*vrs1*.a, *Int-c*.a, *Vrs3*) showing the central grain (Cen) flanked by two lateral grain (Lat). Scale bar, 1 cm. **b**, **c** Six-rowed barley spikes from **b** the current commercial six-rowed genotype model: *vrs1*.a, *Int-c*.a, *Vrs3*.w (662) and **c** with the introduction of *vrs3*: *vrs1*.a, *Int-c*.a, *vrs3* (666), respectively; Scale bar, 1 cm. **d**, **e** Distributions of **d** central and **e** lateral grain width fractions within the 662 genotypes. **f**, **g** Distributions of **f** central and **g** lateral grain width fractions within the 666 genotypes. **h–j** comparison of the mean grain width, grain area and lateral to central grain area ratio between the 662 (*n* = 6) and 666 (*n* = 13) genotype combinations; black bars: 662 genotype, grey bars: 666 genotype. Error bars are ± S.E.D

**VRS3 sequence diversity**. Reciprocal BLAST of rice LOC_Os10g42690 against the cv. Morex barley genome sequence identified Morex contig 5669[12, 13] as containing the putative barley orthologue. Amplicons, generated with overlapping primer pairs were designed to amplify a region of 5815 bases from upstream of the predicted start and downstream of the predicted stop (Supplementary Table 3), were Sanger sequenced using Big-Dye v3.1 (Applied Biosystems) across 22 cultivars representative of two-rowed and six-rowed, winter and spring haplotypes in the region of *VRS3* (Supplementary Data 4). *Vrs3*.w or *Vrs3*.x alleles were discriminated in a further 220 spring and 255 winter cultivars using a single diagnostic KASP allele-specific PCR (primers in Supplementary Data 2). Sequence variation in *VRS3* in landraces and wild barley was obtained from the exome capture sequence data described[19]. Accession geo-reference locations are provided in Supplementary Data 5 and *Vrs3* haplotypes in Supplementary Data 6. A median-joining tree was constructed using PopArt network software (http://popart.otago.ac.nz).

**Gene expression analyses**. For RNA-seq, following germination in 96-well trays, Bowman WT and BW902(*vrs3*.f) seedlings were transferred to 8.89 cm pots arranged in a randomised block design. For each biological rep (*n* = 8, providing >98% statistical power[46]), 20–25 AP stage (AP, 5 mm ± 1 mm spike length) or 10–15 spikes at WA stage (10 mm ± 1 mm spike length) were collected off the main shoot within a constant 3 h window in the light period. Total RNA was extracted with TRIzol Reagent (Invitrogen), treated with DNase (QIAGEN) and purified with RNeasy columns (QIAGEN). Stranded RNAseq libraries were prepared with the NEXTflex™ Rapid Directional RNA-Seq Kit with NEXTflex™ DNA Barcodes (Bioo Scientific, manual v 14.09) using 1 μg total RNA of each sample to generate a library using NEXTflex™ beads. Libraries were normalised, pooled (6 libraries per lane of sequencing) at equimolar concentrations and diluted to 10 pM. Pools were clustered with the HiSeq Rapid PE Cluster Kit v2 (Illumina) and sequenced on a HiSeq2500 (Illumina). Each library pool was run in a single lane as paired-end 2 × 150 bp. Reads were demultiplexed using CASAVA 1.8, allowing a one base-pair

mismatch. For data analysis (Supplementary Fig. 7), raw data were checked for quality using FASTQC 0.11.3 and adaptors trimmed using *Trimmomatic 0.30*[47]. Transcript abundance was calculated using Salmon 0.7.2[48], based on the barley transcript reference[12, 13]. Estimated read counts were summarised to gene level using *tximport 1.2.0*[49]. Consistently lowly expressed genes were filtered out and the trimmed mean of M-values method (TMM) applied to normalise read counts in *edgeR 3.16.5*[50] and then converted to log2-read-counts-per-million (log2CPM). Relationships between mean and variance were estimated and weights for variance adjustments generated using the *voom* function[51] in *limma 3.30.9*[52]. Using a general linear model with genotype and developmental stages as factors, contrast groups identified DE genes[52]. For the contrast groups, an empirical bayes *t* test was used to calculate the *P* values, these were further adjusted to account for multiple testing[53] (Supplementary Note). Gene Ontology enrichment analysis on DE genes (Supplementary Note) identified over-represented GO terms using GOEAST[54]. Selected transcripts were validated by qRT-PCR in independent experiments (Supplementary Fig. 5 and Supplementary Note).

For qRT-PCR, RNA was extracted from developing inflorescences (*n* = 3, 24 plants per rep, according to MIQE guidelines for sample replication[55]) at 1 mm, 2–3 mm and 10 mm length using an RNAeasy Plant Mini Kit (QIAGEN). cDNA was synthesised from 2 μg of RNA using Ready-To-Go You-Prime First-Strand Beads (GE Healthcare Lifesciences). qRT-PCR reactions were run in triplicate (technical reps) on a StepOnePlus (Applied Biosystems) with Universal Probe Library (Roche) hydrolysis probes. Expression was normalised to the *ACT2* or *PDF2* reference using the $2^{-\Delta CT}$ method. Significant differences were calculated using a Student's *t*-test (two-tailed). qPCR primers are listed in Supplementary Table 4 and parameters in Supplementary Note. The 16 tissue RNA-seq Expression Atlas (FPKM values) is available in the BarleyGenes database[b] (Supplementary Note).

**SEM and in situ hybridisation**. For SEM samples, spikes at AP stage were prepared and imaged as described[56]. Spikes collected at early lemma and AP stages

were prepared for in situ hybridization according to Goodall et al.[57]. To synthesise the probe template, 300 bp from the 3'UTR of VRS3 cDNA was cloned into the pCR4-TOPO vector (Invitrogen). Antisense and sense probes were synthesised using primers incorporating the T7 polymerase binding site at the 5′ end (Supplementary Table 5), as described[58]. Hybridisation occurred overnight at 52 °C with 50 ng/µl DIG-labelled RNA probe in hybridisation buffer. Post-hybridisation washes and immunodetection was performed as described previously[58, 59], except slides were incubated for 90 min rather than 4 h with diluted (1:1250) antibody conjugate (anti-DIG-AP, Roche) in BSA wash solution, and then washed three times (15 min each) in BSA wash solution. Sections were photographed under bright-field using the AxioCamHR setup (Zeiss). Empty slide background was colour matched in Photoshop (Adobe) to compare between separate slides.

**Row-type gene pyramids.** Allelic combinations at VRS1, VRS5 and VRS3 were established using KASP allele-specific PCR (LGC Genomics) (Supplementary Data 2) of seedling leaf DNA from 600 BW419*Morex and BW902*Morex F2 plants. 300 F2 plants from each cross were grown to provide a 99% chance of at least one individual from each population having the treble homozygous six-rowed allele combination and provide adequate replication. Lines homozygous for six-row alleles at VRS1 (vrs1.a) and VRS5 (Int-c.a), and vrs3 (mutant allele, n = 12) or Vrs3 (WT allele, n = 6) were selected. Following collection of main spikes at Zadoks growth stage 92 (hard caryopsis), grain number and size for central and lateral grain was determined using a MARVIN grain size analyser (GT Sensorik). ANOVA analysis in Genstat (VSN International) tested for significant differences between vrs1.a, Int-c.a, Vrs3.w and vrs1.a, Int-c.a, vrs3 allelic combinations; assumptions for normality and equal variances were confirmed by residuals. Additional details for field trials are provided in the Supplementary Note.

**Data availability.** VRS3 alleles in cultivated germplasm and vrs3-induced mutant allele sequences are included in Supplementary Data 4 and Supplementary Table 6. RNA-Seq data sets are available from European Nucleotide Archive (ENA), http://www.ebi.ac.uk/ena, under project code PRJEB19243. Exome Capture sequence datasets are published[19]. All the genotypic data described can be requested from the corresponding authors. The authors declare that all the other data supporting the findings of this study are available from the corresponding authors upon request.

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

## Acknowledgements

We acknowledge The Mylnefield Trust and James Hutton Limited for funding H.B. The work would not have been possible without funding for M.Z. from BBSRC Grant No.BB/K01613X/1 to R.W. and S.M.M. We acknowledge the EU Marie Skłodowska Curie FP7-PEOPLE-2013-IEF Project 626291 for supporting M.C.C. R.W., W.T.B.T., R.Z., P.R.-F. and J.R. are grateful for continued support from the Scottish Government Research Program and R.W., S.M.M. and A.J.F. for support from the University of Dundee. S.M.M. acknowledges financial support as a Royal Society of Edinburgh Research Fellow. We would like to thank the following colleagues for helpful discussion, advice, help to access published and unpublished data, and for practical support during this project: Richard Keith, Chris Warden, Marcel Lafos, Vrushali Patil, Nicky Bonar, Linda Milne, Scott McCrimmon, Stephen Wall, Allan Booth, and Ainoa Escrich. We acknowledge the Earlham Institute for expertly generating the RNA-seq data sets.

## Author contributions

R.W., S.M.M., W.T.B.T. and A.J.F.: Conceived and supervised the study. H.B. and A.D.: Conducted the mapping and gene identification, phylogenetic and phenotypic analyses. S.K., H.B. and J.R.: Conducted the diversity analyses. M.C.C. and M.Z.: Conducted the expression analyses. M.Z.: Performed the SEM and the in situ hybridisations. M.C.C., R.Z., W.G. and P.R.-F.: Performed the gene expression quantification and differential expression using RNA-seq and M.C.C. analysed the data. R.W., H.B., S.M.M., M.C.C. and M.Z.: Wrote the manuscript. H.B., M.C.C., R.Z., P.R-F., M.Z., S.M.M. and R.W.: Wrote the Online Methods and Supplementary information. P.R.-F.: Submitted the data to the relevant archives.

## Additional information

**Competing interests:** The authors declare no competing financial interests.

