## [Peer Review File · Nature Communications]

Reviewers' comments:

Reviewer #1 (Remarks to the Author):

In this article by Hazel Bull and colleagues, the authors investigate the genetic regulation of barley inflorescence architecture through analysis of the *vrs3* mutant. The authors show that the causal mutation for the *vrs3* phenotype resides within a gene that encodes a JMJC H3K9me2/3 demethylase, which is supported by extensive analysis of this gene in other mutant lines that fall within the same complementation group. By demonstrating that mutations in *VRS3* influence the expression of other known regulators of barley inflorescence architecture, including *VRS1* and *Int.C/VRS5*, these results elegantly tie together almost a decade's worth of research investigating the genetic control of 2-row versus 6-row inflorescence architecture in barley. The findings presented here are therefore important because they increase our knowledge about an agriculturally important trait for barley, while also expanding our understanding of the biology that underpins inflorescence development in cereals.

The data is technically sound and covers a broad set of experiments and analysis. The RNA-seq analysis, *in situ* hybridisation and results from NILs that combine *vrs3* alleles with *vrs1* and *vrs5* alleles that promote a 6-row phenotype provide strong novelty and insight for future investigation into the regulation of spikelet and floret fertility in barley and other cereals. Hence, the results presented here are important to other researchers investigating the genetic regulation of barley inflorescence development, but also to those investigating inflorescence architecture and fertility traits in other cereals such as wheat, rice and maize.

While this is a well-written manuscript, there are some areas that should be considered by the authors to strengthen the manuscript and broaden its readership.

1. In multiple sections, the manuscript suffers from being written in too brief a format/style. This is possibly due to the selection of journal to which the manuscript was submitted – I encourage the authors and the editors to allow for more expansion in some areas to allow more support for hypotheses and conclusions to be made. Specific examples include:
 - a. More discussion about the conclusion that *vrs3.w* and *vrs3.x* originated from wild species in Israel – it's not clear what analysis led to this conclusion.
 - b. For the RNA-seq analysis, it's not clear why the authors decided to look at the WA stage. The AP stage makes sense based on results shown in Fig. 1, but the rationale for WA is not clear.
 - c. There's a relatively poor description of the hypothesis for focusing on down-regulated genes in the RNA-seq analysis. The authors have written that "H3K9me2/3 is a repressive chromatin mark", but have not explained that *VRS3* is likely to remove these repressive marks. This also links to my comment about the conclusions about epigenetic regulation below.
 - d. I also recommend that the authors include some results from the supplementary note into the main text. For example, the flowering-time data, and the *in situ* results from later stages showing expression of *VRS3* in the rachis (Supp. Fig. 4c-d), as this *in situ* result is very unique and possibly explains the differential expression of genes involved in sugar

metabolism and hormone pathways. Similarly, I recommend the authors include some of the detail about SWEET genes that are currently included in the supplementary notes.

2. My biggest concern with this manuscript regards the strength of language used in the abstract, final paragraph and title about the role for VRS3 in epigenetic regulation of genes, as no actual test for removal of repressive epigenetic marks has been performed in this study. I agree that the genetics, expression analysis and results from rice suggest that the mutations in VRS3 are affecting inflorescence architecture because of failure to remove methyl groups from H3K9, but this has not been investigated biochemically in this study, and so I suggest that the authors tone down their discussion accordingly. I also recommend that the authors change the title to something that is more descriptive about the identity of VRS3 and its link to gene expression, rather than epigenetic regulation.

3. Following on from this, I am intrigued about the role for VRS3 in WT plants, and its regulation of VRS1 and VRS5. In Figure 4, the authors show that VRS1 and VRS5 are already expressed quite high at the AP stage. If VRS3 does remove repressive methyl marks to allow for VRS1 and VRS5 to be expressed, then it is reasonable to expect that expression of these genes would be 'off' or 'very low' at early stages, possibly TM stage. Based on the Supp. Fig 4a, the authors have RNA from TM stage – is VRS1 and VRS5 off or very low at this earlier stage? Inclusion of this result would help support a role for VRS3 in removing repressive marks for developmental regulation of expression for genes important for inflorescence development.

There are also some minor editorial comments

1. Reference 6, a? (First paragraph of main text)
2. 'six-rowed phenotypic penetrance phenotype...' redundant use of word phenotype? (First paragraph of main text)
3. Incorrect reference to Supp. Figure 3 in paragraph discussing allelic diversity. This figure is the tissue specific transcription analysis.

Reviewer: Scott Boden

Reviewer #2 (Remarks to the Author):

This study identifies VRS3, a gene controlling fertility of lateral florets on the barley spike. Mutations in VRS3 lead to a partial 6-row phenotype, where all three florets are fertile (6-row) at some inflorescence internodes, as opposed to only the central floret (2-row). Previously four other VRS genes have been identified and these encode various classes of transcription factor. The authors mapped VRS3 to a region that contains a gene encoding a Jumonji C-type histone3-Lysine demethylase protein (JMJC), which was selected as a candidate for VRS3 based on observed floral morphology defects in rice mutants of the equivalent gene. Gene resequencing of the candidate gene from a large number of allelic mutants supported the hypothesis that the JMJC gene is VRS3, with loss-of-function mutations identified in 31 of 32 mutants examined. Representation of different VRS3 haplotypes were examined in different barleys, including landraces and modern varieties,

including both 2 and 6 row types. RNA-seq was then used to identify misexpressed genes in the VRS3 mutant; genes that might normally be activated as a consequence of histone demethylation. A relatively short list (compared to typical RNA-seq studies) of differentially expressed genes was identified. This included previously identified VRS genes. Additionally, a range of other genes involved in floral development, sugar metabolism or hormone pathways were differentially expressed. Interestingly, introducing the VRS3 gene into a VRS1 six-row barley (typical genotype for six row cultivars) increased grain width and area, suggesting the potential to use VRS3 to increase crop yield.

This is an interesting study of a topic that is of scientific interest and also relevant to crop improvement. It should be of interest to a broad readership, but is especially relevant to cereal geneticists studying floral development and yield in barley, but also wheat, rice, maize and sorghum.

The experiments are sound and the manuscript is clear and easy to read.

There is room for minor improvements, as outlined below.

1. The final paragraph over reaches with the interpretation of the results. There is no evidence that VRS3 directly regulates floral development genes. Effects could be indirect, via other misexpressed genes. Global or targeted chromatin precipitation assays are required to support this statement. Similarly, hormone levels and sugar levels have not been assayed, so it is unclear whether VRS3 influences the balance of either. This final paragraph should be reworded to moderate such statements.
2. Line 62 – what is a partial orthologue?
3. Lines 70, 74. Suggest that a gene annotation is chosen and then consistently applied (e.g. use VRS3 not int-a.1).
4. Line 110. Do you expect VRS3 haplotypes to be under selection in a 2-row type, if VRS1 is epistatic to VRS3? (line 59) Or, are you investigating selection via phenotypes not related to row type? Is there a bias between haplotypes in winter versus spring 6-row barleys? This seems to be the case. If so, is this breeding history or selection?
5. Please define JMJC at first use.

Reviewer #3 (Remarks to the Author):

-What are the major claims of the paper?

VRS3 is a Jumonji C-type H3K9me2/me3 demethylase regulating expression of floral development genes.

Deleterious mutations such as *vrs3.f* induce a phenotypic response primarily by maintaining a repressive chromatin state around the major two-row VRS alleles, restricting transcription and partially phenocopying recessive natural mutations (e.g. *vrs1*) that confer a six-row

phenotype.

Transcriptional responses in *vrs3.f* indicate that plant hormone balance, especially cytokinins and JA, sugar metabolism and stress responses are subsequently involved in promoting lateral floret fertility.

-Are they novel and will they be of interest to others in the community and the wider field?
Yes, specifically the finding related to the JA involvement for spikelet formation in barley appears similar to rice; and hence most likely other grass crops such as wheat or maize.

-If the conclusions are not original, it would be helpful if you could provide relevant references.

n/a

-Is the work convincing, and if not, what further evidence would be required to strengthen the conclusions?

n/a

-Do you feel that the paper will influence thinking in the field?

Yes, related to hormonal balances for spike formation.

And, the relevance for the specific *vrs3.f* allele in combination with other *vrs* alleles showing larger grains in lateral spikelets may have implications for breeding because it shows for the first time the potential of double-*vrs*-mutants for improved breeding.

-Please feel free to raise any further questions and concerns about the paper.

I only have three points here:

1) I found it odd that the discovered gene name of *VRS3* has not been mentioned throughout the entire abstract and title. I recommend to the authors to at least provide the term "putative Jumonji C-type H3K9me2/me3 demethylase" in the abstract.

2) The title as well as the conclusions from authors implies that *VRS3* is an epigenetic regulator that removes repressive H3K9me2/me3 epigenetic marks. Importantly, authors did not provide any molecular evidence for the assumed biological function of *VRS3*, and whether it is indeed a Jumonji C-type H3K9me2/me3 demethylase. The nature of the gene product is purely based on similarity comparisons and gene annotation. Whether this gene really removes epigenetic marks has not been shown. Indirect evidence, however, comes from the transcriptional regulation study (RNAseq) which only suggests the proposed function of *VRS3*. Therefore, I'd urge the authors to significantly tone down their statements in relation to *VRS3* function. For example, in L178-179 authors stated: "We have shown that *VRS3* is a Jumonji C-type H3K9me2/me3 demethylase that is required for barley floral organ development by epigenetically regulating expression of floral development genes." Please rephrase to "We have shown that *VRS3* is a putative Jumonji C-type H3K9me2/me3 demethylase that is required for barley floral organ development by regulating expression of floral development genes."

Or L182-185 "Deleterious mutations such as *vrs3.f* induce a phenotypic response primarily

by maintaining a repressive chromatin state around the major two-row VRS alleles, restricting transcription and partially phenocopying recessive natural mutations (e.g. vrs1) that confer a six-row phenotype.”

Change to “Deleterious mutations such as vrs3.f induce a phenotypic response most likely by maintaining a repressive chromatin state around the major two-row VRS alleles, restricting transcription and partially phenocopying recessive natural mutations (e.g. vrs1) that confer a six-row phenotype.”

Please also reconsider “Epigenetic control” for the title; and the main body of the text (+Suppl.) in the same context.

3) Figure 4A: I’m questioning the labeling of CS in this image. The structures look too small and too young for a CS in comparison with LS. The CS in 4B is more realistic. Also, I do not see the signal in the rachis. There’s overall much more signal in 4A in comparison with the sense probe 4B. What is the signal in the awn of CS in 4B? Improved ISH images of antisense/sense might be useful here.

Reviewer #4 (Remarks to the Author):

This manuscript describes the analysis of a locus for spike morphology VRS3, which controls lateral grain size in barley. The authors used a cross between cv. Barke and a NIL carrying mutation in VRS3 for genetic analysis. By using barley – rice syteny, the authors estimated the VRS3 region to 10.9 cM interval containing 23 gene models and found a candidate HvJMJ706. By resequencing vrs3 mutants previously identified by genetic complementation, allelic variations within HvJMJ706 were identified. The authors also conducted molecular phylogenetic study among wild and cultivated germplasm, expression analysis and estimated the allele effect by making a cross between the mutant and six-rowed cv. Morex. The content of the MS is extensive and I feel that HvJMJ706 may regulate the spike morphology of barley. However, I am not very positive of the manuscript Bull et al. for the publication in Nat Commun by following reasons.

1) Only 108 F2 plants are used to map VRS3. The number of individuals are too small to narrow down the trait to the gene. I also feel that the number of 23 loci is unexpectedly small to correspond to the region of 10.9 cM in barley genome.

2) I understand that the HvJMJ706 may be a strong candidate for VRS3. However, the main evidence is the resequencing results of vrs3 mutants. There is no transformation experiments or fine genetic mapping to support the results. There is still a possibility of other gene models to be VRS3 within the target region. The authors did not mention in the MS that other 22 gene models or other factors in the region were not considered as possible candidates of VRS3.

3) The authors showed that the lateral to central grain size ratio increased from 89% to 93% by introducing a mutant allele of VRS3 to an elite six-rowed cultivar Morex. However, there is no clear evidence that the trait is significantly related to enhance grain yield, which may be a key interest for readers of Nat Commun.

Hazel Bull et al 'VRS3'

Response to reviewers' comments:

First of all we would like to thank the Reviewers for their overall positive and indeed helpful comments about our manuscript. Below we itemise the comments that require attention and our responses. All changes have been incorporated into the revised version of the manuscript and are highlighted in RED text. Individual referees comments are numbered 1-4

Referee 1

1.1a Allelic diversity : 'evidence that Vrs3.w and Vrs3.x originated from wild species in Israel'

We extended the description of the haplotype analysis describing in more detail how we conclude that the origin of the cultivated alleles is likely Israel (p3/4)

1.1b RNA seq: 'rationale for choosing WA stage in RNA seq'

We added a comment that qualified why this stage was chosen in the text (p4, para 3)

1.1c RNA seq: 'poor description of the hypothesis for focusing on down-regulated genes in the RNA-seq analysis'.

Added a sentence that explains this hypothesis based on the function of the WT rice JMJC homologue which is involved in removing repressive chromatin marks.

1.1d Phenotyping: Move flowering time results to main text

We added a sentence that presents our observations on flowering time in vrs3 (p2, para 3)

1.1d In situ: Move in situs of rachis expression to main text

We added rachis expression *in situs* to a reformatted Fig 4. And added a sentence describing these results to the text (p4, para2)

1.1d RNA seq: Move some of SWEET genes detail from supplementary notes to main text

We moved our discussion of the SWEET genes and appropriate references to the main text (p4/5 last/first para)

1.2 Abstract: Tone down use of epigenetic regulation

We adjusted the abstract as requested, adding the terms 'putative Jumonji C-type H3K9me2/me3 demethylase'

1.2 Title: Remove epigenetic regulation from the title and include more description of Vrs3 identity and link to gene expression

We re-titled the manuscript focusing on the genes putative function

1.2 Final paragraph Tone down use of epigenetic regulation

We rewrote the final paragraph as requested.

1.3 qRTPCR at AP stage - Vrs1 and Int-c already highly expressed. Include expression data at triple mound to support role of Vrs3 in removing repressive marks

We have added graphs to supplementary figure 4 illustrating the change in expression of *Vrs3*, *Vrs1* and *Vrs5* across three developmental stages of the Bowman developing inflorescence.

1.4 Paragraph 1: Reference 6, a? (First paragraph of main text)

^a refers to a URL

1.5 Paragraph 1: six-rowed phenotypic penetrance phenotype...

We removed the second 'phenotype' and changed the language in the last two sentences of para 1 as requested

1.6 Allelic diversity: Incorrect reference to Supp. Figure 3 in paragraph discussing allelic diversity.

The ref is Correct – changed to Supplementary Fig 2

Referee 2

2.1 Final paragraph: final paragraph over reaches with the interpretation of the results - no evidence for the direct regulation of floral development genes or hormone & sugar levels

See response to 1.2 above. We re-wrote the final paragraph toning down our conclusions regarding direct epigenetic regulation.

2.2 Introduction: what is a partial orthologue?

We used the term 'partial orthologue' to indicate that the gene has incompletely or 'partially' overlapping (orthologous) functions. However, for clarity we changed the language in the last two sentences of para 1 as requested.

2.3 Phenotyping: Nomenclature (Use of int-a.1 (line 70))

We have removed the reference to *int-a.1* from line 70.

2.4 Allelic diversity: "Do you expect VRS3 haplotypes to be under selection in a 2-row type, if VRS1 is epistatic to VRS3? (line 59)

Or, are you investigating selection via phenotypes not related to row type? Possibly - there are known loci for winter hardiness (winter genepool) and malt quality (spring genepool) on chromosome 1H

Is there a bias between haplotypes in winter versus spring 6-row barleys? Yes – see figure 3a

This seems to be the case. If so, is this breeding history or selection? Possibly

These are good points that we cannot fully answer based on our current data. However to point out these possibilities we added a qualifying statement at the end of p3, para 2.

2.5 Mapping: define JMJC at first use

Done

3.1 Abstract: Discovered gene name missing

We added the gene name to the abstract

3.2 Title: Tone down in relation to VRS3 function

See response to 1.2. We re-titled the manuscript focusing on the genes putative function

3.2 Final paragraph "significantly tone down statements in relation to VRS3 function

Rephrase:

"We have shown that VRS3 is a Jumonji C-type H3K9me2/me3 demethylase that is required for barley floral organ development by epigenetically regulating expression of floral development genes."

"Rephrase:

Deleterious mutations such as vrs3.f induce a phenotypic response primarily by maintaining a repressive chromatin state around the major two-row VRS alleles, restricting transcription and partially phenocopying recessive natural mutations (e.g. vrs1) that confer a six-row phenotype.

We rephrased the final paragraph along the lines suggested by the referee

3.3 In situ: Figure 4a: Labelling of CS in image- structure looks too small and too young

Structures labelled as CS in Figure 4a are oblique sections through the CS. We have added an image of the entire spike in Supplemental Figure 4c which shows these structures are more developed and larger in the upper part of the section.

3.3 In situ: Lack of evidence of signal in rachis

The in situ hybridisation showing VRS3 signal in the rachis was originally in Supplemental Figure 4. We have moved these panels into Figure 4b,c.

3.3 In situ: More signal in 4A compared to 4B; what is the signal in the awn of CS in 4B?

We believe that the greater depth of signal in 4A may represent slide-to-slide variation while the signal in the awn of 4B is background due to the section edge. In situ hybridisations can be challenging to interpret and we recognise that the in situ hybridisations of young spikes presented here are variable. However, we feel that the spikelet signal we detected is consistent with the tissues affected by VRS3 function and given the qPCR and RNA-seq data showing VRS3 expression in these stages of spike development. We offer to remove the VRS3 in situ panels from the younger spike tissue if the reviewer requests.

4.1 Mapping: Small population size, 23 loci small compared to 10.9cM region

We extended our description of the genetic mapping – explaining how it was performed in two stages – crude then fine mapping, resolving the region to a recombinogenic 2.1cM region containing 23 gene models.

4.2 Mapping: ‘Absence of transformation’ and ‘No mention of the other 22 loci not being considered candidates’

We argue strongly that the identification of 31 independent deleterious mutations in a single well-conserved gene is ample proof that the gene is causal without need for transformation experiments. Such evidence would be accepted by (almost) all referees. Such unequivocal evidence also obviates the need to look at the other 22 genes.

4.3 Treble homozygous combination: No clear evidence that trait is significantly related to enhanced grain yield

The referee is correct as there is no significant increase in spike grain yield by incorporating *vrs3.f*, and we have added an additional graph to Supplementary Figure 6 to illustrate this. However, we do not make this claim. We stress that the phenotypes we observe are in lateral grain width and area, an important processing character as small lateral grains over modify during malting and ‘unifying’ grain size is therefore a desirable trait.

To accommodate these modifications we have extended and renumbered the references and modified Figure 4 and Supplementary Figure 4.

REVIEWERS' COMMENTS:

Reviewer #2 (Remarks to the Author):

The authors have responded to comments raised in the initial review.

A few minor points that the authors might consider:

Line 90. With regard to reviewer 4, comment 2, I accept the authors logic that the genetic evidence of multiple alleles with mutations in the same gene is extremely strong evidence that VRS3 corresponds to the JMJC demethylase gene. But, given discussion a bout linkage to other genes potentially driving selection of haplotypes in different gene pools, I am curious about what the other closely linked genes are. Can these be listed in a supplementary table?(or did I miss where these are listed?). Also, the differential expression of linked genes was filtered out of the expression comparison between the near-isogenic lines. Maybe these data should be included in the same table?

Line 229: "If VRS3 acts solely by regulating...other VRS genes". The RNA seq analysis identified many candidates other than VRS genes. Some of these genes could be mentioned here. This raises the question of whether some differentially expressed genes are regulated by VRS3 or by the downstream VRS genes. So, do any of the other VRS genes affect grain width in a six-row background? (VRS1 loss of function).

Reviewer #3 (Remarks to the Author):

see my comments in the attachment

"Barley *SIX-ROWED SPIKE 3 (VRS3)* is a putative Jumonji C-type H3K9me2/me3 demethylase that represses lateral floret fertility by facilitating expression of the major VRS genes"

Bull H., et al.

REV#3 My own points:

Ad 3.1) Abstract: Discovered gene name missing

-I'm happy authors considered changing the title and abstract accordingly. I'm wondering whether VRS in the title "major VRS genes" should be italicized?!

Ad 3.2) Title: Tone down in relation to VRS3 function

3.2) Final paragraph "significantly tone down statements in relation to VRS3 function"

-I'm satisfied with the changes made.

Ad 3.3) In situ: Figure 4a: Labelling of CS in image- structure looks too small and too young.

-Authors state that these are oblique sections. This may indeed explain why we see LS and CS in the same image.

Following this, the odd nipple-like structure on top of the LS dome (which I found quite strange previously; because there usually is no "nipple" on the spikelet/floret dome!) may in fact be the glume tip of the CS behind. Please check for that in Kirby & Appleyard (1984, 2nd ed.) for LP stage fig 5.9; lateral view. Here you find a similar "nipple" on top of the LS dome, suggesting that the VRS3 signal is NOT in the lemma but rather in the glumes of CS and LS. If so, please re-phrase legend of fig4 (L556) and the main text.

Since there is no implication/evidence for lemma expression of VRS3 (unless you have another ISH image); the awn phenotype statement should be toned down or re-phrased (L199-200).

Additional comments:

-Authors may carefully evaluate the text whether they mean the gene (capitalized, italicized), the protein (capitalized) or the locus (lower case, italicized) name of VRS3. There are a few inconsistencies in the entire text incl. supplement.

For example:

L084, correct; ...at the vrs3(italicized) locus.

L153, correct; ...that VRS3(not-italicized) may... (I think here you mean protein function.)

-L147, Delete "...and when ... determined"; not required here.

Editor comment:

I realize that the points you raised were fairly minor. However, reviewer #4 raised some more serious concerns regarding the evidence that variation in VRS3 was indeed responsible for the floret phenotype. I was hoping you may be able to give your opinion on this issue.

4.1 Mapping: Small population size, 23 loci small compared to 10.9cM region

Authors' response: We extended our description of the genetic mapping - explaining how it was performed in two stages - crude then fine mapping, resolving the region to a recombinogenic 2.1cM region containing 23 gene models.

I'm completely fine with the authors' point.

4.2 Mapping: 'Absence of transformation' and 'No mention of the other 22 loci not being considered candidates'

Authors' response: We argue strongly that the identification of 31 independent deleterious mutations in a single well conserved gene is ample proof that the gene is causal without need for transformation experiments. Such evidence would be accepted by (almost) all referees. Such unequivocal evidence also obviates the need to look at the other 22 genes.

I'm completely fine with the authors' point. One additional point authors have not been mentioning here, but in the text, is the fact that many of such 32 mutants have been shown to be allelic in a previous work (see L100). So, no doubt about the low genetic resolution; this is the gene!

4.3 Treble homozygous combination: No clear evidence that trait is significantly related to enhanced grain yield

Authors' response: The referee is correct as there is no significant increase in spike grain yield by incorporating *vrs3.f*, and we have added an additional graph to Supplementary Figure 6 to illustrate this. However, we do not make this claim. We stress that the phenotypes we observe are in lateral grain width and area, an important processing character as small lateral grains over modify during malting and 'unifying' grain size is therefore a desirable trait.

I'm completely fine with the authors' point. The claim "to enhance grain yield" was never made.

Reviewer #2 (Remarks to the Author):

2.1 Line 90. With regard to reviewer 4, comment 2, I accept the authors logic that the genetic evidence of multiple alleles with mutations in the same gene is extremely strong evidence that VRS3 corresponds to the JMJC demethylase gene. But, given discussion about linkage to other genes potentially driving selection of haplotypes in different gene pools, I am curious about what the other closely linked genes are. Can these be listed in a supplementary table? (or did I miss where these are listed?). Also, the differential expression of linked genes was filtered out of the expression comparison between the near-isogenic lines. Maybe these data should be included in the same table?

(Supplementary Table 1 now lists the other genes within the mapped vrs3 interval to show closely linked genes. Differentially expressed genes within the vrs3.f introgression were already listed in Supplementary Data 1b (column heading 'Introgression Region?'))

2.2 Line 229: "If VRS3 acts solely by regulating...other VRS genes". The RNA seq analysis identified many candidates other than VRS genes. Some of these genes could be mentioned here. This raises the question of whether some differentially expressed genes are regulated by VRS3 or by the downstream VRS genes. So, do any of the other VRS genes affect grain width in a six-row background? (VRS1 loss of function).

(We have re-worded the final discussion to widen the possible involvement of other differentially expressed genes in improving the lateral to central grain width ratio. Further work, probably with single and double mutant combinations, will be required to fully address the effect of other VRS genes on the lateral to central grain width ratio).

Reviewer 3:

3.2 *In situ*: Figure 4a: Labelling of CS in image- structure looks too small and too young. -Authors state that these are oblique sections. This may indeed explain why we see LS and CS in the same image. Following this, the odd nipple-like structure on top of the LS dome (which I found quite

strange previously; because there usually is no “nipple” on the spikelet/floret dome!) may in fact be the glume tip of the CS behind. Please check for that in Kirby & Appleyard (1984, 2nd ed.) for LP stage fig 5.9; lateral view. Here you find a similar “nipple” on top of the LS dome, suggesting that the VRS3 signal is NOT in the lemma but rather in the glumes of CS and LS. If so, please re-phrase legend of fig4 (L556) and the main text. Since there is no implication/evidence for lemma expression of VRS3 (unless you have another ISH image); the awn phenotype statement should be toned down or re-phrased (L199-200).

(We have added another panel to the ISH in Supplementary Figure 4 which shows signal in the lemma; however we have removed the phrase “...and lemmas...” from the main text. We have consulted the reference referred to by referee#2 and agree that the “nipple” signal may reflect a glancing section of the central spikelet. Thus, we have relabelled Figure 4a (“...VRS3 signal in the developing glumes (gl) of the lateral spikelet (LS, in middle plane) and potential signal () from the glume of the adjacent central spikelet (CS) and...” We have also removed the emphasis of lateral awn growth from the Results subheadings.*

Regarding the second point, regardless of the spatial expression dynamics of VRS3 which we recognise are challenging to interpret, the “...ectopic awn/lemma and floret phenotypes...” are undeniable (Figure 1 and Supplementary Note 1) and have relevance to the DE of jasmonate-related genes and their orthologue’s function in controlling rice floral organ development. Accordingly, this phrase is unaltered.)

Additional comments:

3.3 -Authors may carefully evaluate the text whether they mean the gene (capitalized, italicized), the protein (capitalized) or the locus (lower case, italicized) name of VRS3. There are a few inconsistencies in the entire text incl. supplement.

For example:

L084, correct; ...at the vrs3(italicized) locus.

L153, correct; ...that VRS3(not-italicized) may... (I think here you mean protein function.)

(We revised the ms. and checked what we mean each time the nomenclature is used as follows. VRS – protein, VRS – gene, Vrs – wild type allele, vrs – mutant allele

-L147, Delete “...and when ... determined”; not required here.

(We have deleted this phrase as suggested).